# Preparation and NH_3_ Gas-Sensing Properties of Double-Shelled Hollow ZnTiO_3_ Microrods

**DOI:** 10.3390/s20010046

**Published:** 2019-12-19

**Authors:** Pi-Guey Su, Xiang-Hong Liu

**Affiliations:** Department of Chemistry, Chinese Culture University, Taipei 111, Taiwan

**Keywords:** double-shelled hollow, ZnTiO_3_ microrods, NH_3_ gas sensor, room-temperature

## Abstract

A novel double-shelled hollow (DSH) structure of ZnTiO_3_ microrods was prepared by self-templating route with the assistance of poly(diallyldimethylammonium chloride) (PDDA) in an ethylene glycol (EG) solution, which was followed by calcining. Moreover, the NH_3_ gas-sensing properties of the DSH ZnTiO_3_ microrods were studied at room temperature. The morphology and composition of DSH ZnTiO_3_ microrods films were analyzed using scanning electron microscopy (SEM), transmission electron microscopy (TEM) and X-ray diffractometry (XRD). The formation process of double-shelled hollow microrods was discussed in detail. The comparative gas-sensing results revealed that the DSH ZnTiO_3_ microrods had a higher response to NH_3_ gas at room temperature than those of the TiO_2_ solid microrods and DSH ZnTiO_3_ microrods did in the dark. More importantly, the DSH ZnTiO_3_ microrods exhibited a strong response to low concentrations of NH_3_ gas at room temperature.

## 1. Introduction

ZnO and TiO_2_ films have been extensively studied for use in sensing, but traditional ZnO and TiO_2_ gas sensors, which are based on ZnO and TiO_2_ films, can typically only be used at temperatures from 300 to 500 °C [1,2]. Additionally, selectivity is also an important property of metal oxide-based sensors. Several designs of sensors’ construction were proposed for achieving the selectivity of these sensors, such as directly doping small amounts of noble metals (Au, Pd, and Pt), a suitable filter containing the noble metal catalysts method, and nano-carbon-based composite materials film [3,4,5,6]. ZnO and TiO_2_ are well known inorganic photocatalysts, so that ultra-violet (UV) irradiation has been used to reduce the operating temperature of these sensors [7,8,9,10]. ZnO–TiO_2_ binary oxide systems have a better photocatalytic performance than single systems [11]. The ZnO-TiO_2_ binary oxide system had three compounds: they are ZnTiO_3_ (cubic, hexagonal), Zn_2_TiO_4_ (cubic, tetragonal) and Zn_2_Ti_3_O_8_ (cubic) [12,13,14,15,16,17,18]. ZnTiO_3_ has attracted particular interest because of its potential for use in the photocatalysis of the degradation of organic pollutants, and in adsorption and microwave devices [18,19,20]. Numerous reports have revealed that ZnTiO_3_ has favorable photocatalytic properties in visible light [21,22,23,24], favoring its use for gas sensors at room temperature [25]. Yadav et al. [25] fabricated a ZnTiO_3_ nanopowders film using a physicochemical method for sensing liquefied petroleum gas (LPG) at room temperature. Ippolito et al. [26] fabricated an acetone gas sensor that was made of ZnTiO_3_ nanoarrays using a hydrothermal method. The limit of detection (LOD) of this sensor under light and at 350 °C was 10 ppb.

One-dimensional (1D) nanostructured materials such as tubes, wires, belts, and rods have great potential for use in gas sensors, not only because of their excellent optical, electrical, and mechanical properties, but also because of their efficiency and activity sites, which are caused by their high porosity and large surface area [18,27]. Hollow micro/nanostructures have also attracted enormous interest because of their many hollow cavities, which make the surface area of hollow structures significantly greater than that of their solid counterparts [28]. In recent decades, many methods have been used to prepare ZnTiO_3_ nanopowders, such as the conventional solid-state reaction, the molten salt method, the sol-gel method, the chemical bath deposition and the hydrothermal method [13,14,15,16,29,30,31]. The physicochemical properties of ZnTiO_3_ nanopowders depend on their morphology, the size of the crystallites, and the crystallographic structure. Recently, Chi et al. [18] fabricated pristine solid ZnTiO_3_ microrods, and You et al. [32] fabricated a reduced grapheme oxide decorated solid ZnTiO_3_ microrod composite using the polyvinylpyrrolidone (PVP)-assisted sol-gel method for use in the photo-degradation of rhodamine B. However, no attempt has been made to prepare a double-shelled hollow (DSH) structure of ZnTiO_3_ microrods and to study their NH_3_ gas-sensing properties at room temperature.

Ammonia (NH_3_) is known to be highly hazardous to the environment and the human body because of its high toxicity. Accordingly, the fabrication of NH_3_ gas sensors that can be used at room temperature, with a high response and a low production cost, has attracted much attention. In this work, a novel room-temperature NH_3_ gas sensor with a high sensitivity, based on novel DSH ZnTiO_3_ microrods, was fabricated. DSH ZnTiO_3_ microrods were prepared using a self-templating approach, by heating sol-gel derived Zn-Ti glycolates with poly(diallyldimethylammonium chloride) (PDDA) in an ethylene glycol (EG) solution and then calcining. This method is very simple, has a low production, and does not involve heterogeneous coating, so it can be easily scaled up for the fabrication of sensors. A plausible process for the fabrication of DSH structures is proposed. X-ray diffraction (XRD), scanning electron microscopy (SEM), transmission electron microscopy (TEM), selected-area electron diffraction (SAED), and energy dispersive X-ray (EDX) analysis were used to characterize the composition and morphologies of the DSH ZnTiO_3_ microrods. The NH_3_-sensing properties of DSH ZnTiO_3_ microrods at room temperature, including the sensing response, sensing linearity, selectivity, response/recovery times, repeatability, stability, and sensing mechanism, were also studied.

## 2. Experimental Methods

### 2.1. Materials

The following chemicals were used as received without further purification: titanium (IV)-ethylhexanoate (Ti[(OOCCH(CH_2_)_4_(CH_3_)_2_)]_4_ (TE, Alfa Aesar), zinc acetate dehydrate (Zn(OAc)_2_·2H_2_O; Sigma-Aldrich, St. Louis, MO, USA), ethylene glycol (EG, J. T. Baker), poly (diallyldimethylammonium chloride (PDDA, molecular weight (Mw) = 200,000~350,000, Aldrich).

### 2.2. Fabrication of Gas Sensors Based on DSH ZnTiO_3_ Microrods and Measurement of Their Sensing Properties

1.25 g TE, 0.25 g Zn(OAc)_2_·2H_2_O and 0.3 mL PDDA were added to a 10 g EG solution, which was then stirred at 190 °C for 1.5 h. An as-prepared precursor solution (PDDA-Zn-Ti-glycolates rods precursor) was drop-coated on an alumina substrate with interdigitated electrodes (IDE). The system was then calcined at 500 °C for 4 h at a heating rate of 5 °C min^−1^ for decomposing the matrix polymer and organic groups, and for oxidizing and crystallizing the Zn-Ti-glycolates. Figure 1a shows a picture of the structure of the as-prepared NH_3_ gas sensors. The preparation and characterizations of the TiO_2_ solid microrods were completed according to our previous report [33].

The electrical and NH_3_ gas-sensing characteristics of the DSH ZnTiO_3_ microrods were measured using a bench system, as shown in Figure 1b. The volume of the bench system is 18 L. A Direct current (DC) mode was used to measure the resistance of the as-prepared sensors. A power supply (GW, PST-3202) applied a fixed 5 V to the sensor circuit. A DAQ device (NI, USB-6218) was used to measure the resistance of the sensor in various concentrations of NH_3_ gas. A standard 1000 ppm NH_3_ gas in N_2_ gas (Shen Yi Gas Co., Taiwan) was used to prepare the required various NH_3_ gas concentrations. The desired various gas concentrations were prepared by diluting the known volume of standard NH_3_ gas with dry air, and were calibrated by a standard gas sensor system (Dräger, MiniWarn). A fan was used to disperse the testing gases inside the bench system and was purged with air. All experiments were measured at room temperature (about 23.0 ± 1.5 °C) and the relative humidity at 45% RH. The response (S) of the sensors was calculated according to Equation (1):
(1)$$S\;(\%)=\frac{\left(R_{\mathrm{air}}-R_{\mathrm{gas}}\right)}{R_{\mathrm{air}}}\times 100\%.$$
R_air_ and R_gas_ are the electrical resistances of the sensor in the air and testing gas at the exposure time of 300 s, respectively.

### 2.3. Characterization of DSH ZnTiO_3_ Microrods

The composition and morphologies of the DSH ZnTiO_3_ microrods film coated on an alumina substrate were investigated using X-ray diffraction (XRD) using Cu K_α_ radiation (Shimadzu, Lab XRD-6000), scanning electron microscope (SEM, JEOL JSM-5310), transmission electron microscopy (TEM, JEM-1400; JEOL, Tokyo, Japan), selected-area electron diffraction (SAED) and energy dispersive X-ray (EDX) analysis.

## 3. Results and Discussion

### 3.1. Characteristics of DSH ZnTiO_3_ Microrod Film

#### 3.1.1. XRD Characterization of DSH ZnTiO_3_ Microrods

Figure 2a,b presents the XRD spectra of the DSH ZnTiO_3_ microrods that were calcined at 500 °C for 4 h and without calcining, respectively. The reflections at (220), (311), (400), (511) and (440) agree closely with the cubic crystal phase of ZnTiO_3_. No peak that corresponded to the TiO_2_, ZnO or zinc titanates, all of which are associated with other stoichiometries, was observed. These results are consistent with the literature [18,32]. The XRD results further verified that the formation and crystallinity of the as-prepared ZnTiO_3_ at temperatures as low as 500 °C were attributable mainly to the short diffusion paths of metal ions during the heat treatment of the PDDA-Zn-Ti-glycolates rods as the precursor in the polyol processing [14]. Additionally, the diffraction peaks of the DSH ZnTiO_3_ microrods without calcining were unobvious and broad, indicating a poor crystalline structure (Figure 2b).

#### 3.1.2. SEM and TEM Analyses of Morphology of DSH ZnTiO_3_ Microrod Film

Figure 3 presents the SEM images of the partially DSH ZnTiO_3_ microrod film that was prepared by the self-templating approach, which was followed by calcining. Figure 3a reveals the presence of clearly regular microrods, which had aggregated into bundles. The high-magnification SEM image of the DSH ZnTiO_3_ microrods (Figure 3b) shows that the microrods were linked together in an interconnected porous network structure. The length and diameter of the DSH ZnTiO_3_ microrods were about 3~6 μm and 0.25~0.45 μm, respectively. The microrod structure had round and narrow tips and open ends (indicated by arrows). Figure 4 presents the TEM images of the DSH ZnTiO_3_ microrods that were synthesized by PDDA-assisted self-templating. The low-magnification TEM image of a single DSH ZnTiO_3_ microrod (Figure 4a) shows a partially hollow structure at its end. The higher-magnification TEM image of the selected area in Figure 4a indicates that the ZnTiO_3_ microrods had a double-shelled structure (indicated by arrows) (Figure 4b). The shell was thin, and the internal diameter of the hollow rod was about 0.23 μm. The high-resolution TEM (HRTEM) image of the DSH ZnTiO_3_ microrods indicates that the lattice spacing of the adjacent lattice planes was about 0.27 nm, consistent with the (220) crystal plane of cubic ZnTiO_3_ (Figure 4c). The SAED pattern (inset in Figure 4c) confirms that the microrods comprised cubic ZnTiO_3_, consistent with the relevant XRD results. To further investigate the composition of the DSH ZnTiO_3_ microrods, EDX elemental mapping and an elemental analysis (Figure 4e) were conducted. The EDX elemental mapping (Figure 4d) suggested the presence of Zn, Ti, and O only in the DSH ZnTiO_3_ microrods. The EDX elemental analysis (Figure 4e) revealed an atomic ratio of Zn to Ti of close to 1:1, revealing a close match with the stoichiometric composition.

Figure 5 presents a plausible synthesis of the DSH ZnTiO_3_ microrods. First, chain-like sol-gel-derived Zn-Ti-glycolates were formed by heating in an EG solution. EG is well known to serve as a complexing agent in the formation of Zn-glycolate and Ti-glycolate from Zn^2+^ and Ti^4+^ ions (path (1)) [34], respectively. The Zn-glycolate was then intercalated with Ti-glycolate, forming chain-like Zn-Ti-glycolates, similar to those described elsewhere [34,35,36]. Then (path (2)), during the sol-gel process, PDDA was adsorbed on the surface of these chain-like Zn-Ti-glycolates by electrostatic attraction at a high temperature, forming hollow PDDA-Zn-Ti-glycolates rods as a precursor. In the cooperative assembly process, PDDA exhibited the dual action of protecting and etching. PDDA molecules were wrapped outside the chain-like Zn-Ti-glycolates in the initial stage of heating, protecting them in a stable shell. Since the PDDA (with quaternary amines) was coated on the surfaces of chain-like Zn-Ti-glycolates, the increase in the amount of counterions (OH^−^) nearby increased the local alkalinity and facilitated etching accordingly. Hollow PDDA-Zn-Ti-glycolates rods were thus formed as a precursor [28,37]. Additionally, if many of the surface Zn-Ti-glycolates rods became covered by PDDA, the penetration of etching species into the interior of the Zn-Ti-glycolates rods was prevented, and no hollow structure was formed. Finally (path (3)), the hollow PDDA-Zn-Ti-glycolates rods as the precursor underwent a post-calcination treatment at a high heating rate (5 °C min^−1^). The outmost PDDA-Zn-Ti-glycolates layer with a limited thickness was concentrated; the outermost ZnTiO_3_ shell was degraded and oxidized; and the inner PDDA-Zn-Ti-glycolates layer was contracted. Thereafter, the outermost ZnTiO_3_ shell separated from the shrinking internal PDDA-Zn-Ti-glycolates layer. Subsequently, the inner ZnTiO_3_ shell formed in the same way until the organic groups in the precursor PDDA-Zn-Ti-glycolates burned out [38]. As a result, DSH ZnTiO_3_ microrods were formed.

### 3.2. NH_3_ Gas-Sensing Properties of DSH ZnTiO_3_ Microrod Film

Figure 6 presents the responses (S) of the TiO_2_ solid microrods, ZnTiO_3_ powders, DSH ZnTiO_3_ microrods, and DSH ZnTiO_3_ microrods in the dark, to 100 ppm of NH_3_ gas at room temperature. The response (S) values for the TiO_2_ solid microrods, ZnTiO_3_ powders, DSH ZnTiO_3_ microrods, and DSH ZnTiO_3_ microrods in the dark were 16.17, 17.75, 45.32, and 21.07, respectively. The DSH ZnTiO_3_ microrods exhibited a stronger response (S) than those of the TiO_2_ solid microrods and ZnTiO_3_ powders, and the DSH ZnTiO_3_ microrods in the dark. This result may be attributable to the fact that the DSH ZnTiO_3_ microrods exhibited a higher surface area, larger pores, and a greater total pore volume than the TiO_2_ solid microrods and ZnTiO_3_ powders did. Moreover, the fact that ZnTiO_3_ exhibited a higher response in the visible light than in the dark may be attributable to the fact that the DSH ZnTiO_3_ microrods had a good photocatalytic activity in the visible light [39,40]. Figure 7a presents the dynamic responses (S) of the DSH ZnTiO_3_ microrods to various concentrations of NH_3_. They exhibited a response (S) of 5.1%, even to a low NH_3_ testing concentration of 1 ppm. The limit of detection (LOD) was estimated at the lower calibration point of 1 ppm by considering a S/N of 3. The LOD was 0.45 ppm. Figure 7b presents the linear dependence of the response (S) of the DSH ZnTiO_3_ microrods on the concentration of NH_3_ gas. The sensitivity ($\frac{\Delta S}{\Delta C}$) is obtained from the slope of the linear sensing curve. The linear sensing properties in the ranges of 1 to 20 ppm and 20 to 200 ppm of NH_3_ gas differed. The sensitivity at 5 to 150 ppm of NH_3_ gas was larger than that at 150 to 300 ppm, and a rapid decrease in the slope was observed from 20 to 200 ppm of the NH_3_ gas. This result was related to the synergistic effect of the surface area and the photocatalytic activity of the DSH ZnTiO_3_ microrods. As the concentration of NH_3_ increased to 20~200 ppm, the number of active sites for adsorption decreased, causing a rapid decline in the slope. Figure 8 plots the real-time resistance of the DSH ZnTiO_3_ microrods to 5 ppm of NH_3_ over time. The response time (ResT_90_) and recovery (RecT_90_) times are calculated as the time taken for the resistance of the sensor to change by 90% of its maximum change after the exposing time of the NH_3_ gas at 300 s. The response (ResT_90_) and recovery (RecT_90_) times of the DSH ZnTiO_3_ microrods were 93 and 363 s, respectively. Figure 9 plots the response and recovery times as a function of the NH_3_ gas concentration. The recovery time increased with an increasing NH_3_ gas concentration. The rather long recovery time was attributable to the hollow interior cavities of the DSH ZnTiO_3_ microrods. The sensor also exhibited a good reversibility. Figure 10 plots the effect of the ambient humidity on the response (S) of the DSH ZnTiO_3_ microrods. The response (S) of the DSH ZnTiO_3_ microrods decreased with an increase in the ambient humidity, with measurements at testing concentrations of NH_3_ of 5 ppm. This result was reasonable because the physisorbed water occupied the active sites of the DSH ZnTiO_3_ microrods. Figure 11 plots the results concerning the interfering effects of CO, H_2_, NO_2_, NO, and SO_2_ gases on the DSH ZnTiO_3_ microrods. These interfering gases may be regarded as having unobvious interference effects with NH_3_ at 100 ppm. However, NO_2_ and NO gases detectably interfered with NH_3_ at less than 5 ppm. Figure 12 plots the long-term stability of the DSH ZnTiO_3_ microrods. The mean response (S) of the DSH ZnTiO_3_ microrods to 50 ppm and 5 ppm NH_3_ gas for 68 days were 49.11 and 17.34, respectively. The response drift for 68 days was calculated as the relative standard deviation (RSD). The RSD for the DSH ZnTiO_3_ microrods to 50 ppm and 5 ppm NH_3_ gas were 6.5% and 8.3%, respectively. The relative standard deviation (RSD) of the response (S) of the DSH ZnTiO_3_ microrods to 50 ppm NH_3_ gas was 6.0%, indicating its favorable repeatability. The NH_3_ gas-sensing properties of the presented NH_3_ sensor was compared with those of sensors in the literature, as shown in Table 1 [41,42,43,44,45,46]. The DSH ZnTiO_3_ microrods had the lowest detection limit for sensing NH_3_ gas at room temperature.

### 3.3. Electrical Properties and NH_3_ Gas-Sensing Mechanism of DSH ZnTiO_3_ Microrod Film

Figure 13 plots the real-time resistance of the DSH ZnTiO_3_ microrod film as a function of time for various concentrations of NH_3_. The resistance of the DSH ZnTiO_3_ microrod film herein was reduced by exposure to NH_3_ gas (electron-donating). Accordingly, the prepared DSH ZnTiO_3_ microrod film had the electrical property of an n-type semiconductor. Therefore, the changes in resistance of the DSH ZnTiO_3_ microrod film by exposure to NH_3_ gas have been suggested from the reports of Hieu et al. [47], Gupta et al. [48], and Shi et al. [49], as illustrated in Equations (2)–(4) [47,48,49]:
O_2 (g)_ + e^−^ O_2 (ads),_(2)
NH_3(g)_ NH_3(ads),_(3)
4 NH_3(ads)_ + 3 O_2 (ads)_^−^ 2 N_2_ + 6 H_2_O + 6 e^−^.(4)


First, the atmospheric oxygen adsorbed electrons from the conduction band of the surface of the DSH ZnTiO_3_ microrod film, forming O_2(ads)_^−^ (Equation (2)). Then, the adsorption of electron-donating NH_3_ gas molecules interacts with pre-adsorbed oxygen ions (O_2(ads)_^−^) and releases electron carriers into the n-type DSH ZnTiO_3_ microrod film, causing its electrical resistance to decrease, while increasing the concentration of NH_3_ gas (Equations (3) and (4)). From all of the above, two main effects are proposed to explain why the DSH ZnTiO_3_ microrod film had the strongest response (S). First, the DSH ZnTiO_3_ microrods had hollow cavities, and their consequently large surface area favored NH_3_ gas adsorption. Second, the fact that the response of the DSH ZnTiO_3_ microrod film in light was stronger than in the dark is directly related to its photocatalytic activity. In the presence of sunlight, electrons were photo-excited from the valence band to the conduction band of the DSH ZnTiO_3_ microrods, and these photo-generated electrons at the surface of the DSH ZnTiO_3_ microrods were then transferred to the adsorbed oxygen, increasing the adsorbed oxygen ions, favoring the chemisorption of NH_3_, and thereby improving the response of the DSH ZnTiO_3_ microrods [28,29,33,34].

## 4. Conclusions

Novel DSH structures of ZnTiO_3_ microrods were fabricated via a PDDA-assisted self-templating approach in an EG solution and were then calcined. The synergistic surface-protecting and core-etching of chain-like Zn-Ti-glycolates by PDDA explains the formation of double-shelled hollow structures of the ZnTiO_3_ microrods. The ZnTiO_3_ microrods exhibited a strong response to low concentrations of NH_3_ gas at room temperature, including a good sensitivity (5.1%) at 1 ppm NH_3_, a good linearity (Y = 1.5575 X + 7.5526; R^2^ = 0.9068) at 1~20 ppm NH_3_, a fast response time (93 s), a good repeatability, a good reversibility, a high selectivity, and a good long-term stability (at least 68 days).

## Figures and Tables

**Figure 1 sensors-20-00046-f001:**
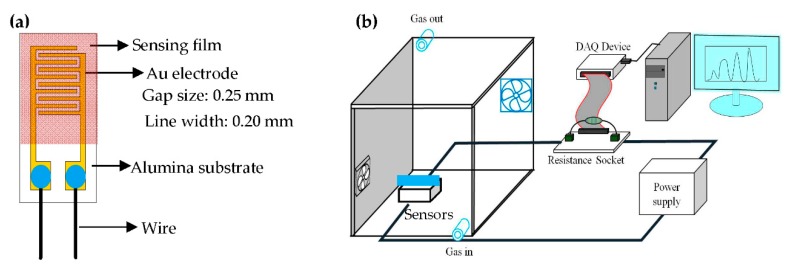
(**a**) The structure of the NH_3_ gas sensor and (**b**) the measurement system for testing the gas sensors.

**Figure 2 sensors-20-00046-f002:**
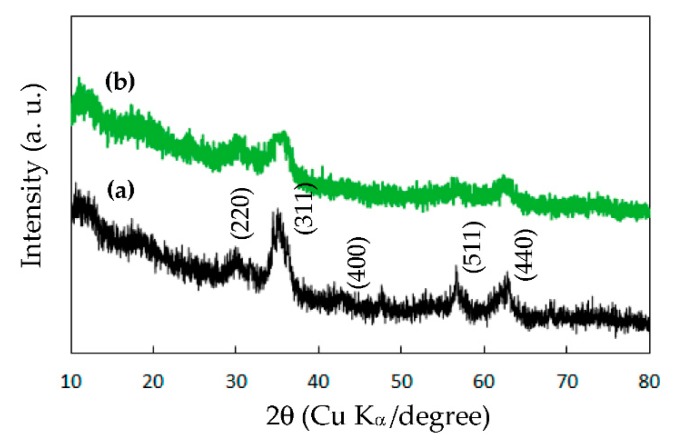
The XRD patterns of the DSH ZnTiO_3_ microrods that were fabricated via the self-templating approach, followed (**a**) by calcining and (**b**) without calcining.

**Figure 3 sensors-20-00046-f003:**
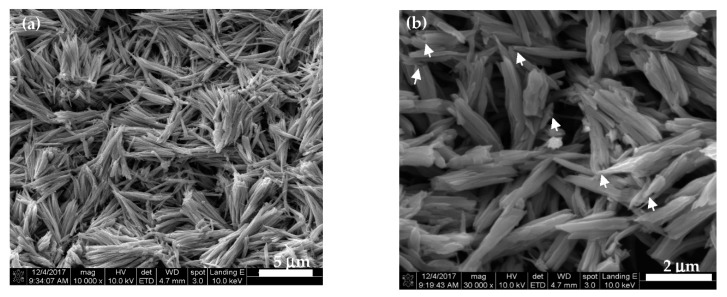
The SEM images of the DSH ZnTiO_3_ microrods that were fabricated via the self-templating approach, followed by calcining: (**a**) low magnification and (**b**) high magnification.

**Figure 4 sensors-20-00046-f004:**
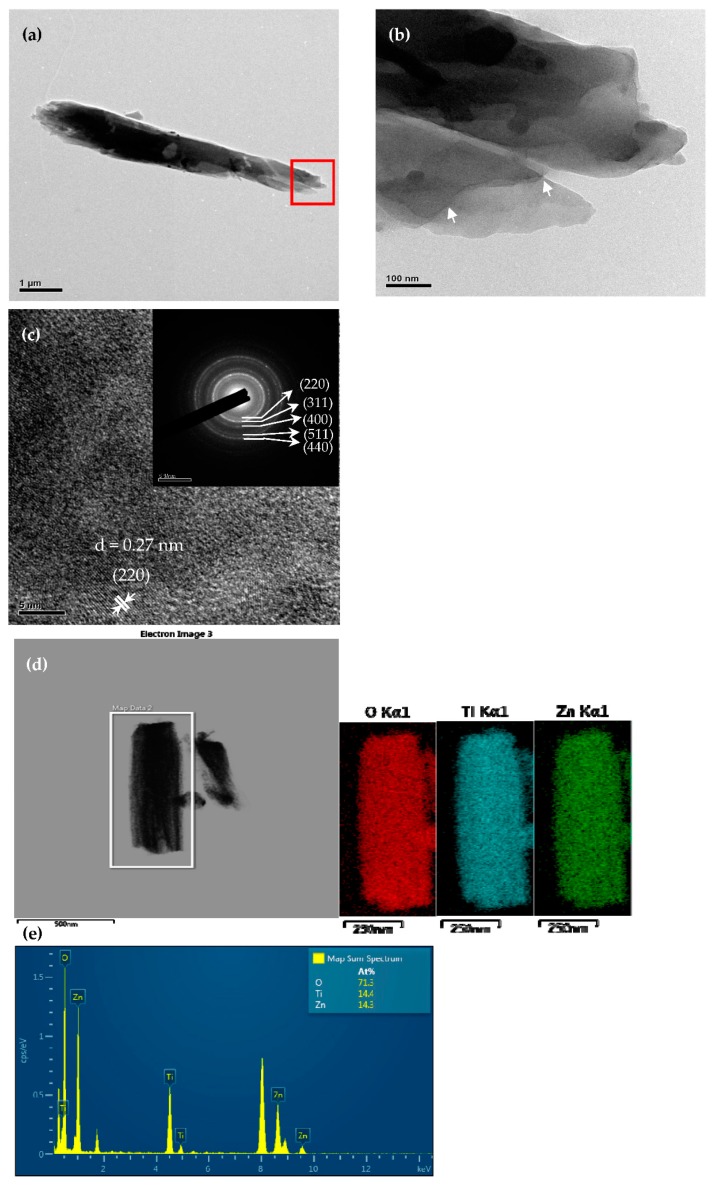
The TEM images of the DSH ZnTiO_3_ microrod: (**a**) low-magnification TEM, (**b**) high-magnification TEM, (**c**) HRTEM (the inset is the corresponding SAED pattern), (**d**) EDX elemental maps and (**e**) EDX spectrum.

**Figure 5 sensors-20-00046-f005:**
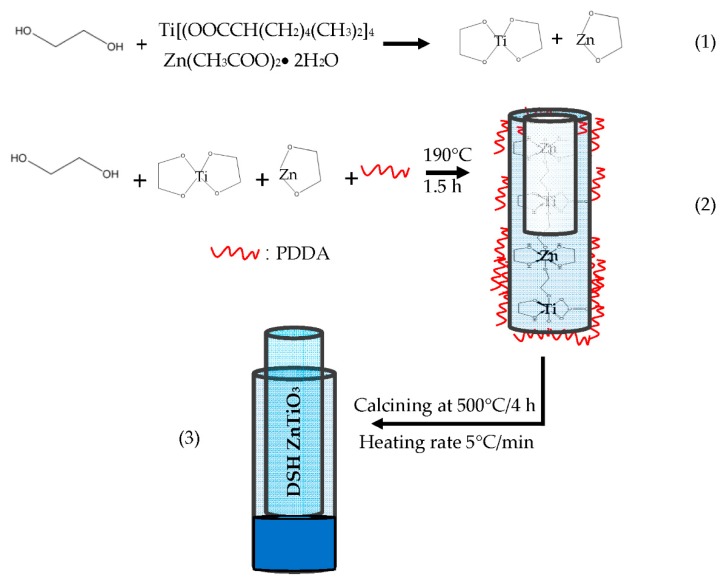
Schematic illustration of the fabrication of DSH ZnTiO_3_ microrods by the self-templating approach.

**Figure 6 sensors-20-00046-f006:**
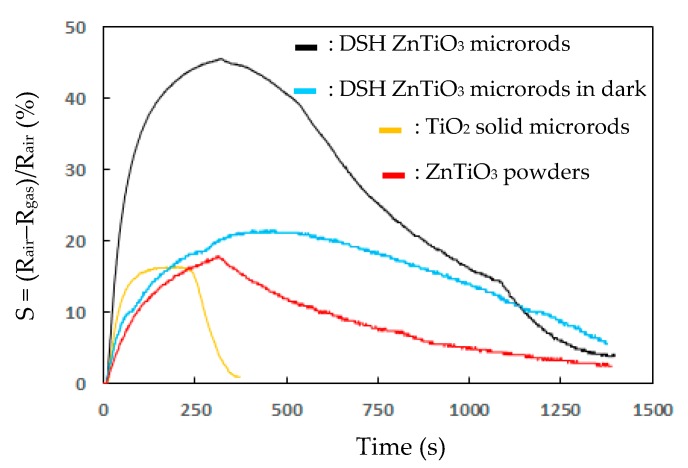
The response (S) of intrinsic TiO_2_ microrods, ZnTiO_3_ powders, DSH ZnTiO_3_ microrods, and DSH ZnTiO_3_ microrods in the dark, in 100 ppm NH_3_ gas at room temperature.

**Figure 7 sensors-20-00046-f007:**
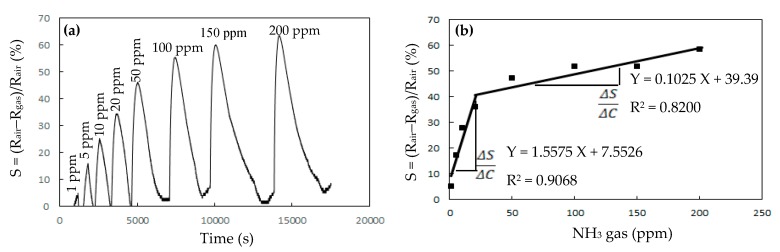
(**a**) The response (S) of the DSH ZnTiO_3_ microrods to various concentration of NH_3_ gas at room temperature, and (**b**) the linear dependence of the response (S) of the DSH ZnTiO_3_ microrods on the concentration of NH_3_ gas at room temperature. The sensitivity ($\frac{\Delta S}{\Delta C}$) is determined from the slope of the linear curve.

**Figure 8 sensors-20-00046-f008:**
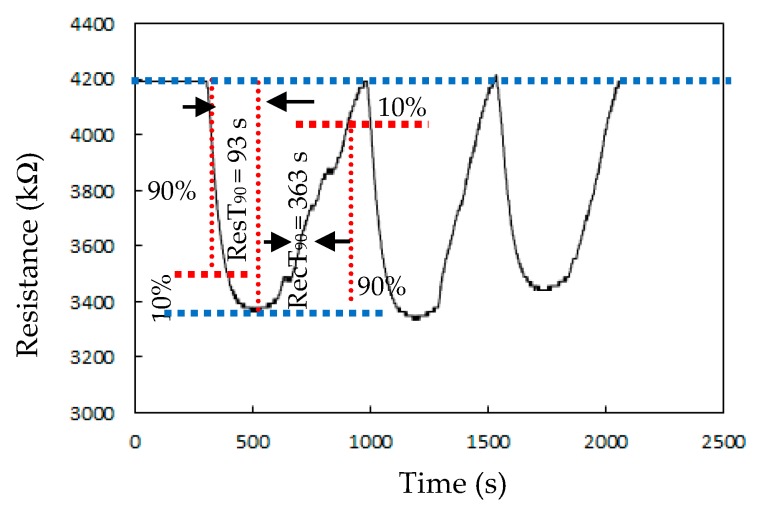
The response and recovery of the DSH ZnTiO_3_ microrods to 5 ppm NH_3_ at room temperature.

**Figure 9 sensors-20-00046-f009:**
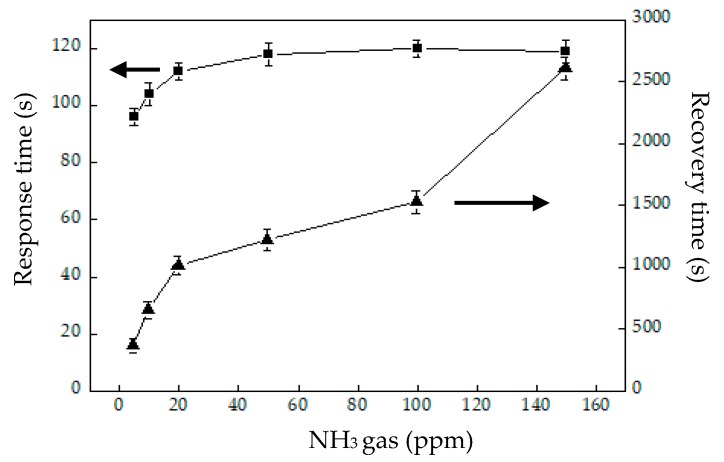
The response and recover times vs. the NH_3_ gas concentration. (■) response time and (▲) recovery time.

**Figure 10 sensors-20-00046-f010:**
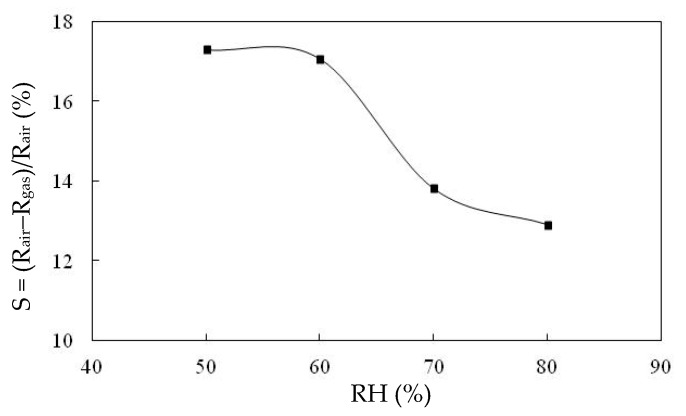
The effect of the ambient humidity on the response (S) of the DSH ZnTiO_3_ microrods.

**Figure 11 sensors-20-00046-f011:**
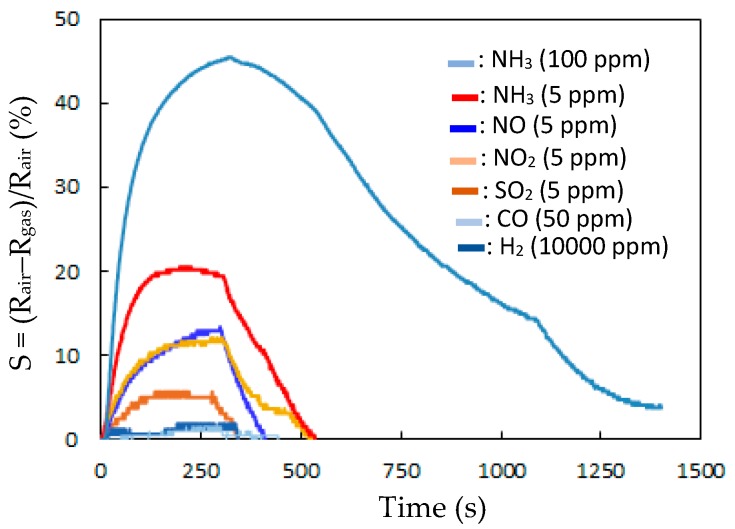
The response (S) of the DSH ZnTiO_3_ microrods to various gases.

**Figure 12 sensors-20-00046-f012:**
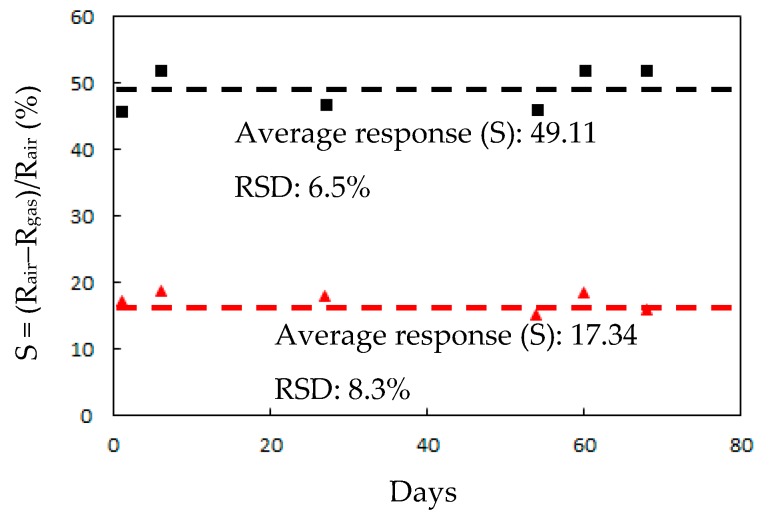
The long-term stability of a NH_3_ gas sensor based on DSH ZnTiO_3_ microrods (dotted line as the average response (S) value). (■) 50 ppm and (▲) 5 ppm NH_3_ gas.

**Figure 13 sensors-20-00046-f013:**
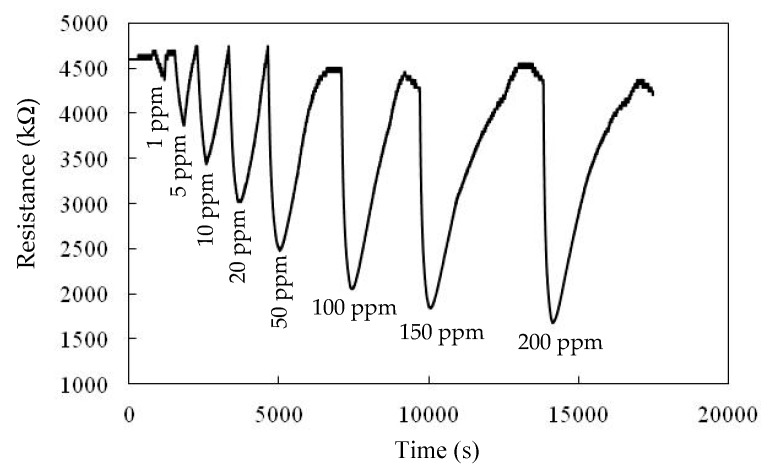
The real-time resistance of DSH ZnTiO_3_ microrods as a function of time (s) toward different NH_3_ concentrations from 1 to 200 ppm.

**Table 1 sensors-20-00046-t001:** Comparison of the performance of the NH_3_ gas sensor developed herein with the literature.

Sensing Material	Operating Temperature (°C)	Detection Limit (ppm)	Response/Recovery Time (s)	References
TiO_2_	25	5	34/90	[41]
TiO_2_ microspheres/RGO	25	5	-/-	[42]
TiO_2_/RGO	25	-	55/-	[43]
ZnO/NiO	25	15	20/90	[44]
ZnO/Pd	200	30	198/334	[45]
Pd NPs/TiO_2_ MRs/RGO	25	2.4	420/3000	[46]
DSH ZnTiO_3_ microrods	25	1	93/363	This work

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
