# Peer review of "Preparation and NH3 Gas-Sensing Properties of Double-Shelled Hollow ZnTiO3 Microrods"

_sensors, 2019, doi:10.3390/s20010046_

Round 1
Reviewer 1 Report
This work fabricated ZnTiO3 microrods that have double-shelled hollow structures and successfully showed NH3 sensing properties of the fabricated microrods. After polishing it based on the following minor critiques, this paper may be fit for publication in Sensors. Therefore, I would like to recommend that this paper needs minor revision to be published in Sensors.
1. The authors claimed that the fabricated ZnTiO3 microrods have double-shelled hollow structures, but the only experimental evidence of it is Fig. 4a,b, which is not very clear and only shows partial double-shelled hollow structures. I would like to recommend the authors to present additional experimental data (example: multiple TEM images that clearly show DSH structures). Otherwise, the authors might want to tone down their claim (double-shelled hollow structures).
2. In Fig. 6, NH3 gas-sensing properties of the DSH ZnTiO3 microrods were compared to those of the TiO2 solid microrods. Why did you use the TiO2 solid microrods (why didn't you use ZnTiO3 powders or ZnTiO3 solid microrods) as a reference?
3. Line 9: Novel double-shelled hollow (DSH) "structure" of ZnTiO3 microrods "were" ...
4. Fig.1(a): Some characters are missing.
5. Fig. 10: Please align gas names in the horizontal axis.
Author Response
Reply to reviewer 1’s comments:
1. The authors claimed that the fabricated ZnTiO3 microrods have double-shelled hollow structures, but the only experimental evidence of it is Fig. 4a,b, which is not very clear and only shows partial double-shelled hollow structures. I would like to recommend the authors to present additional experimental data (example: multiple TEM images that clearly show DSH structures). Otherwise, the authors might want to tone down their claim (double-shelled hollow structures).
Response:
We thank the reviewer’s suggestion. The “partially” was added in lines 164 and 172.
2. In Fig. 6, NH3 gas-sensing properties of the DSH ZnTiO3 microrods were compared to those of the TiO2 solid microrods. Why did you use the TiO2 solid microrods (why didn't you use ZnTiO3 powders or ZnTiO3 solid microrods) as a reference?
Response:
We thank the reviewer’s suggestion. The response of the ZnTiO3 powders was studied and the results were added in Fig. 6. The explanation was added in lines from 306 to 313.
3. Line 9: Novel double-shelled hollow (DSH) "structure" of ZnTiO3 microrods "were" ...
Response:
The "structure" of ZnTiO3 microrods "were" ... was changed to "structure" of ZnTiO3 microrods "was" ... as shown in line 9.
4. Fig.1(a): Some characters are missing.
Response:
The mistake was corrected as shown in line 111.
5. Fig. 10: Please align gas names in the horizontal axis.
Response:
The original Fig. 10 was changed to new Fig. 11 and the plot was changed to that the sensing curve of each gas rather than a bar chart to clearly show the gas names.

Reviewer 2 Report
This paper reports on ZnTiO3 microrods and their NH3 sensing capabilities. Below are issues that need to be attended to for clarification and better discussion. I suggest publication of this work after major revision.
From the abstract…”which was followed by calcining, and for NH3 gas-sensing at room-temperature.” I believe this statement needs correction. From the introduction…”nanostructured materials such as (tubes, wires, belts, and rods) have….” Remove the brackets What is the type of “pair of comb-like electrodes” used? Fix all figures quality. Under SEM discussion..” The microrod structure had round and narrow tips and open ends…” I can’t see this from the image, provide an image which clearly shows this Authors only provided characterization from XRD, SEM and TEM, yet they attribute their NH3 sensing on the “synergistic effect of the surface area and photocatalytic activity”….”This result is attributable to that DSH ZnTiO3 microrods exhibited a higher surface area, larger pores and a greater total pore volume than the TiO2 solid microrods. Moreover, ZnTiO3 exhibited very good photocatalytic activity in visible light so the response of the DSH ZnTiO3 microrods therein was stronger than in the dark”… . Provide the relevant characterization (surface area and photocatalytic activity analysis) to demonstrate these characteristics. Also, under the sensing there is now an additional material which is TiO2 solid microrods, yet the authors did not provide any synthesis and characterization of this material for proper comparison. Authors should add this material from the material synthesis and also characterizations. What are the response values for sensors in fig 6? “The rather long recovery time was attributable to the hollow interior cavities of the DSH ZnTiO3 microrods”…. This statement is not complete, please elaborate. Explain why the NH3 has high response even in the presence of the interfering gases? What is the response drift after 68 days? On fig 6 the authors are comparing responses of the sensors to 100 ppm NH3, then they use 5ppm NH3 to demonstrate the repeatability characteristics and also the response and recovery time, and to demonstrate the effect of humidity…the authors further continued to demonstrate the response of the sensor in the presence of interfering gases using 100 ppm NH3…then they continued to demonstrate long term stability for 68 days to 50 ppm NH3….this is confusing, please explain how are you choosing these concentrations?
Author Response
Reply to reviewer 2’s comments:
1. From the abstract…”which was followed by calcining, and for NH3 gas-sensing at room-temperature.”
Response:
The suggestion was corrected as shown in lines 11 and 12.
2. I believe this statement needs correction. From the introduction…”nanostructured materials such as (tubes, wires, belts, and rods) have….” Remove the brackets
Response:
The brackets were deleted as shown in line 42.
3. What is the type of “pair of comb-like electrodes” used?
Response:
It is interdigitated electrodes and the explanation was corrected as shown in line 82.
4. Fix all figures quality.
Response:
We thank the reviewer’s suggestion. We checked and corrected the font size of the x and y-axis of all figures in the manuscript.
5. Under SEM discussion..” The microrod structure had round and narrow tips and open ends…” I can’t see this from the image, provide an image which clearly shows this
Response:
We are sorry that the SEM can not clearly show the microrod structure had round and narrow tips and open ends because that only shows partial double-shelled hollow structures were fabricated as shown by arrows in figure. From the TEM results (Figs. 4(a) and 4(b)), the double-shelled hollow structure was observed.
6. Authors only provided characterization from XRD, SEM and TEM, yet they attribute their NH3 sensing on the “synergistic effect of the surface area and photocatalytic activity”….”This result is attributable to that DSH ZnTiO3 microrods exhibited a higher surface area, larger pores and a greater total pore volume than the TiO2 solid microrods. Moreover, ZnTiO3 exhibited very good photocatalytic activity in visible light so the response of the DSH ZnTiO3 microrods therein was stronger than in the dark”… . Provide the relevant characterization (surface area and photocatalytic activity analysis) to demonstrate these characteristics.
Response:
We thank the reviewer’s suggestion. The response of the ZnTiO3 powders was studied and the results were added in Fig. 6. The explanation was added in lines from 306 to 310. The DSH ZnTiO3 microrods exhibited a stronger response (S) than those of TiO2 solid microrods and ZnTiO3 powders. Moreover, the DSH ZnTiO3 microrods under visible light exhibited a stronger response (S) than that in dark. Additionally, many papers have confirmed the photocatalytic activity of ZnTiO3.
7. Also, under the sensing there is now an additional material which is TiO2 solid microrods, yet the authors did not provide any synthesis and characterization of this material for proper comparison. Authors should add this material from the material synthesis and also characterizations.
Response:
The preparation and characterizations of the TiO2 solid microrods have been studied in our previous work. The explanation about the preparation and characterizations of the TiO2 solid microrods was added in lines 85 and 86.
8. What are the response values for sensors in Fig. 6?
Response:
The response values for sensors in Fig. 6 was added and the explanation was added in lines 308 and 309.
9. “The rather long recovery time was attributable to the hollow interior cavities of the DSH ZnTiO3 microrods”…. This statement is not complete, please elaborate.
Response:
The results about the response and recovery time Vs. gas concentration were added to show the effect of the NH3 gas concentration on the recovery time and showed as Fig. 9. The explanation was added in lines 330 and 331.
10. Explain why the NH3 has high response even in the presence of the interfering gases?
Response:
The major reasons are that the sensor operated at room-temperature and had good photocatalytic activity in visible light.
11. What is the response drift after 68 days?
Response:
The response drift for the sensor was calculated as the relative standard deviation (RSD). The explanation was added in lines from 342 to 344.
12. On fig 6 the authors are comparing responses of the sensors to 100 ppm NH3, then they use 5 ppm NH3 to demonstrate the repeatability characteristics and also the response and recovery time, and to demonstrate the effect of humidity…the authors further continued to demonstrate the response of the sensor in the presence of interfering gases using 100 ppm NH3…then they continued to demonstrate long term stability for 68 days to 50 ppm NH3….this is confusing, please explain how are you choosing these concentrations?
Response:
The 100 ppm NH3 was chosen for studying the response of the different sensors because of showing the obvious deviation in response for the sensors that were made of various sensing materials. The effect of the NH3 gas concentration on the response and recovery times was added as Fig. 9 and the explanation was added in lines 330 and 331. The humidity exhibited higher interference on the sensor under low concentration of NH3 gas than that in high concentration of NH3 gas condition. Therefore, the 5 ppm NH3 was chosen for studying the effect of the humidity on the sensor. We also show the long-term stability of the sensor at 5 ppm NH3 in Fig. 12. The explanation was added in lines from 341 to 344.

Reviewer 3 Report
Pi-Guey Su et al. reported the NH3 gas-sensing properties of double-shelled hollow ZnTiO3 microrods. The authors have characterized the sample extensively to arrive at the parameters which directly affect the sensing. A few more clarifications are, however, needed before the manuscript can be considered suitable for publication. Novelty missing since similar work reported earlier in (ACS Appl. Mater. Interfaces 2019, 11, 32, 29255-29267). Overall, the paper has many weaknesses that need to be overcome. A list of other comments that need to be addressed follows:
An introduction is poorly written, please include the scope and limitations of the ZnTiO3-based sensors, their advantages, and disadvantages since metal oxides are widespread material, and it is hard to find the novelty in the paper. BC Yadav et al. has reported the liquefied petroleum gas sensing using ZnTiO3 Nanostructures, they reported the ppm level of LPG with fast response, recovery speed, and reproducibility. How the presented work is better than this work. It looks repeatability. Selectivity and operation temperature are a crucial issue of metal oxide-based gas sensors. How Does this material address this issue? It is better to mention in the introduction of how the selectivity could be Enhanced? Some review articles to cite in the presentation: Sensors 2019, 19(6), 1285, Sensors and Actuators B 262(2018)758–770, Microchim Acta (2018) 185: 213, Sensors 2018, 18(5), 1456 Experimental Part: Design of experiment missing: why the author chooses reaction temp and time as 500 ℃ for 4h. What are the results for lower and high temperatures? If crystallization is the main reason, please provide XRD before and after annealing. What is the purpose of the two-stage sintering process? How Gas concentration has been calibrated? Please explain in detail. At room temperature, humidity is the primary interference, and it is essential to study gas sensing properties under humid conditions. Please provide response and recovery time Vs. gas concentration with an error bar. Figure 7(d) What is the lower detection limit according to power-law dependence? Figure 6and 7(a): You have not measured steady-state response properties. Please re-measure all data. Figure 10: Please include the selectivity plot with the sensing curve of each gas rather than a bar chart. The quality of some figures is inferior and needs to be enhanced. It is better to check and correct the font size of the x and y-axis of all figures in the manuscript. It should be the same.
Author Response
Reply to reviewer 3’s comments:
1. Novelty missing since similar work reported earlier in (ACS Appl. Mater. Interfaces 2019, 11, 32, 29255-29267).
Response:
We thank the reviewer’s suggestion. The paper was added as Ref. 26 and the explanation was added as in lines from 39 to 41.
2. Selectivity and operation temperature are a crucial issue of metal oxide-based gas sensors. Some review articles to cite in the presentation: Sensors 2019, 19(6), 1285, Sensors and Actuators B 262(2018)758–770, Microchim Acta (2018) 185: 213, Sensors 2018, 18(5), 1456.
Response:
The explanation about the selectivity and the references were added in lines from 26 to 29.
3. Why the author chooses reaction temp and time as 500 ℃ for 4h? What are the results for lower and high temperatures?
Response:
The hollow PDDA-Zn-Ti-glycolates rods were formed as precursor underwent post-calcination treatment at a high heating rate (5°C min-1). The hollow PDDA-Zn-Ti-glycolates rods were not formed at high temperature. The explanation was shown in lines from 267 to 274.
4. If crystallization is the main reason, please provide XRD before and after annealing.
Response:
The XRD before and after annealing was added in Fig. 2. The explanation was added in lines from 136 to 138.
5. What is the purpose of the two-stage sintering process?
Response:
The path 2 was used to form the precursor of hollow PDDA-Zn-Ti-glycolates rods. The path 3 was used to form the DSH ZnTiO3 microrods. The clear explanation was shown in lines from 257 to 274.
6. How Gas concentration has been calibrated?
Response:
A standard 1000 ppm NH3 in N2 gas was used. The concentration of NH3 gas in the chamber was calibrated using a calibrated standard gas sensor system (Dräger, MiniWarn). The explanation was shown in lines 94 and 95.
7. At room temperature, humidity is the primary interference, and it is essential to study gas sensing properties under humid conditions.
Response:
The effect of the humidity on the sensor was shown in Fig. 10. The explanation was shown in lines from 333 to 337.
8. Please provide response and recovery time Vs. gas concentration with an error bar.
Response:
The results about the response and recovery time Vs. gas concentration were added and showed as new Fig. 9. The explanation was added in lines 330 and 331.
9. What is the lower detection limit according to power-law dependence?
Response:
The explanation about the LOD was added in lines 317 and 318.
10. Figure 6and 7(a): You have not measured steady-state response properties.
Response:
We thank the reviewer’s suggestion. In this work, the sensors had good repeatability using dynamic mode. It is well known that the dynamic mode is used for quantitatively determination in analytical chemistry. Moreover, we clearly defined the ∆R (Rair-Rgas) calculations at that point of the exposure time of 300 s in this work and the explanation was added in experimental section in lines 99 and 100. We think the performance of the sensor in this work may be comparable to other works.
11. Figure 10: Please include the selectivity plot with the sensing curve of each gas rather than a bar chart.
Response:
The original Fig. 10 was changed to new Fig. 11 and the plot was changed to that the sensing curve of each gas rather than a bar chart to clearly show the gas names.
12. The quality of some figures is inferior and needs to be enhanced. It is better to check and correct the font size of the x and y-axis of all figures in the manuscript.
Response:
We thank the reviewer’s suggestion. We checked and corrected the font size of the x and y-axis of all figures in the manuscript.

Round 2
Reviewer 2 Report
I'm happy with the rest of comments, authors have addressed them to satisfactory. However, authors did not clearly address the following comment:
Authors only provided characterization from XRD, SEM and TEM, yet they attribute their NH3 sensing on the “synergistic effect of the surface area and photocatalytic activity”….”This result is attributable to that DSH ZnTiO3 microrods exhibited a higher surface area, larger pores and a greater total pore volume than the TiO2 solid microrods. Moreover, ZnTiO3 exhibited very good photocatalytic activity in visible light so the response of the DSH ZnTiO3 microrods therein was stronger than in the dark”… . Provide the relevant characterization (surface area and photocatalytic activity analysis) to demonstrate these characteristics. It raises a lot of doubts to me to attribute the good response on properties that you can’t even give evidence of. Yet it is mentioned that: “Additionally, many papers have confirmed the photocatalytic activity of ZnTiO3”…have authors performed such studies on their material or are these studies that they are mentioning based on their previous works on the very same material.
Otherwise, authors need to withdraw such statements if they didn't conduct these analysis themselves and provide other relevant reasons for the observed high response.
Author Response
Authors only provided characterization from XRD, SEM and TEM, yet they attribute their NH3 sensing on the “synergistic effect of the surface area and photocatalytic activity”….”This result is attributable to that DSH ZnTiO3 microrods exhibited a higher surface area, larger pores and a greater total pore volume than the TiO2 solid microrods. Moreover, ZnTiO3 exhibited very good photocatalytic activity in visible light so the response of the DSH ZnTiO3 microrods therein was stronger than in the dark”… . Provide the relevant characterization (surface area and photocatalytic activity analysis) to demonstrate these characteristics. It raises a lot of doubts to me to attribute the good response on properties that you can’t even give evidence of. Yet it is mentioned that: “Additionally, many papers have confirmed the photocatalytic activity of ZnTiO3”…have authors performed such studies on their material or are these studies that they are mentioning based on their previous works on the very same material. Otherwise, authors need to withdraw such statements if they didn't conduct these analysis themselves and provide other relevant reasons for the observed high response.
Response:
We thank the reviewer’s suggestion. We deleted the explanation the “synergistic effect of the surface area and photocatalytic activity” in line 18.

Reviewer 3 Report
All the comments addressed properly, the manuscript is ready for publication
Author Response
All the comments addressed properly, the manuscript is ready for publication.
Response:
We thank the reviewer’s suggestion.
